# 'We do not rush to the hospital for ordinary wounds *(suḷu tuvāla)*': A qualitative study on the early clinical manifestations of cutaneous leishmaniasis and associated health behaviours in rural Sri Lanka

Sonali Dinushika Gunasekara[1], Nuwan Darshana Wickramasinghe[1], Suneth Buddhika Agampodi 📧[2,3]*, Manoj Sanjeewa Fernando[4], Kosala Gayan Weerakoon[5], Chandani Liyanage[6], Lisa Dikomitis[7‡], Thilini Chanchala Agampodi[1‡]

1 Department of Community Medicine, Faculty of Medicine and Allied Sciences, Rajarata University of Sri Lanka, Saliyapura, Anuradhapura, Sri Lanka, 2 Section of Infectious Diseases, Department of Internal Medicine, School of Medicine, Yale University, New Haven, Connecticut, United States of America, 3 International Vaccine Institute, Seoul, Republic of Korea, 4 Department of Health Promotion, Faculty of Applied Sciences, Rajarata University of Sri Lanka, Mihintale, Anuradhapura, Sri Lanka, 5 Department of Parasitology, Faculty of Medicine and Allied Sciences, Rajarata University of Sri Lanka, Saliyapura, Anuradhapura, Sri Lanka, 6 Department of Sociology, Faculty of Arts, University of Colombo, Colombo, Sri Lanka, 7 Kent and Medway Medical School, University of Kent and Canterbury Christ Church University, Canterbury, United Kingdom

‡ These authors are joint senior authors on this work.
* suneth.agampodi@yale.edu

## Abstract

### Background

Knowledge of early clinical manifestations, people's perceptions and behaviours is crucial in preventing and controlling neglected tropical diseases (NTDs). Cutaneous leishmaniasis is an NTD that causes skin lesions and affects millions worldwide. Delayed healthcare-seeking behaviour leading to prolonged treatment periods and complications is rife among people with cutaneous leishmaniasis. This study examined the patient-reported early clinical manifestations of cutaneous leishmaniasis, local interpretations and associated health behaviours within the socio-cultural context of rural Sri Lanka.

### Methodology/principal findings

We conducted a qualitative study among people with cutaneous leishmaniasis in three rural communities in the Anuradhapura district, Sri Lanka. Participants' experiences were explored through a study-bespoke participant experience reflection journal and in-depth interviews. We analysed the data using a narrative-thematic approach. The study included 30 people with cutaneous leishmaniasis (12 females and 18 males) aged between 18 and 75 years. We identified four major themes during the analysis: 1) patient-reported early clinical manifestations of cutaneous leishmaniasis, 2) local interpretations of the early skin lesion(s), 3) associated actions and behaviours, and 4) the time gap between the initial

**Data Availability Statement:** All relevant data are within the manuscript and its supporting information files.

**Funding:** This research was carried out as part of the ECLIPSE programme funded by the National Institute for Health and Care Research (NIHR https://www.fundingawards.nihr.ac.uk/award/NIHR200135) (NIHR200135) using UK aid from the UK Government to support global health research (LD, SBA, TC, NDW, KGW, CL, SDG). The views expressed in this article are those of the authors and not necessarily those of the NIHR or the UK Department of Health and Social Care. The funders had no role in study design, data collection, and analysis, decision to publish, or preparation of the manuscript.

**Competing interests:** The authors have declared that no competing interests exist.

notice of symptoms and seeking healthcare for cutaneous leishmaniasis. Early clinical manifestations differed among the participants, while the majority misinterpreted them as a mosquito/ant bite, pimple, wart, eczema, macule, or worm infestation. Participants undertook different context-specific self-management actions to cure cutaneous leishmaniasis. We identified an average time gap between the notice of symptoms and the first visit to the healthcare facility ranging from three to twelve months.

## Conclusions/significance

Diverse early clinical manifestations, local interpretations, and associated behaviours of people with cutaneous leishmaniasis have led to a substantial delay in healthcare-seeking. The study sheds light on the importance of understanding the manifestations of NTDs within the social context. Our findings will inform designing context-specific health interventions to improve awareness and healthcare-seeking in cutaneous leishmaniasis in rural settings.

### Author summary

Sri Lanka records a high annual incidence rate of leishmaniasis. Early diagnosis and prompt treatment for cutaneous leishmaniasis are crucial to reduce complications and infection transmission and to facilitate a speedy recovery. Delayed healthcare-seeking in cutaneous leishmaniasis hinders effective disease management leading to a significant burden for the affected people and the healthcare systems. Therefore, this study explored how people with cutaneous leishmaniasis describe and interpret early symptoms of the disease and behave in a certain way leading to a substantial time gap between the notice of symptoms and seeking healthcare. Our study findings are important for three primary purposes; 1) for designing awareness-raising and public health campaigns related to cutaneous leishmaniasis for the general public, 2) for healthcare professionals to conduct an early and accurate clinical diagnosis of cutaneous leishmaniasis and initiate prompt treatment following laboratory confirmation, leading to much-improved disease management and 3) for policymakers to revisit and tailor the national and regional level guidelines and programmes related to leishmaniasis toward successful prevention, early diagnosis, effective treatment and control of the disease.

## Introduction

Leishmaniasis is a neglected tropical disease caused by the *Leishmania* parasite transmitted to humans through the bite of an infected female sandfly of the genera of *Phlebotomus* or *Lutzomyia* [1]. It is endemic in many countries in Asia, Africa, the Americas, and the Mediterranean [2]. Leishmaniasis is categorised into several forms based on a spectrum of clinical manifestations; cutaneous leishmaniasis (CL) with lesions only in the skin, mucocutaneous leishmaniasis (MCL) that causes mucosal tissue destruction, and visceral leishmaniasis (VL) that leads to disseminated visceral infection and internal body organ damage [2]. Global estimates show CL is the widespread form of leishmaniasis, leading to 0.6 to one million new cases annually [3]. Variations of CL include localised cutaneous leishmaniasis (LCL) characterised by ulcerative skin lesions at the sandfly bite site, and diffuse cutaneous leishmaniasis (DCL) with multiple non-ulcerative nodules [4].

A typical CL lesion generally occurs with a small papule or erythema, which may resemble a non-specific insect bite [4–6]. Ulceration of the initial lesion and progressive enlargement of the nodules or plaques are common in CL. These lesions are usually found on the exposed areas of the body, mainly the upper and lower limbs and face [5,7,8]. Clinical manifestations of CL can vary primarily with the *Leishmania* species, host behaviour, and immune response [9]. Therefore, atypical presentations of CL are often reported across endemic regions, including Brazil [10], Turkey [11], Iran [12], Tunisia [12], Pakistan [8], India [13], and Sri Lanka [13,14].

Spontaneous healing of CL within 12–18 months with a lifelong scar is possible [5]. Effective treatment is crucial to speed recovery, prevent scarring and dissemination, and reduce infection transmission [15]. CL is a curable disease, but currently, there are no vaccines or prophylactic drugs to prevent it [16]. Therefore, early diagnosis and prompt treatment are pivotal in disease control [16]. Diversity of the early clinical manifestations of CL leads to delays in seeking healthcare by people [9,17,18] and misdiagnosis by healthcare professionals [19,20], often hindering early case detection.

Leishmaniasis is endemic in Sri Lanka, where there is a higher annual incidence (2217 CL cases in 2020) in comparison to some countries in Asia, such as Nepal (12 CL cases in 2020), Bhutan (2 CL cases in 2021) and Thailand (zero CL cases in 2020 and 2021) [21]. CL is the most prevalent form of leishmaniasis in Sri Lanka [22]. Leishmaniasis is currently under the country's national surveillance system for communicable diseases [23], and evidence shows the existence of the disease within the country since the early 20th century [24,25]. *Leishmania donovani* is considered to be the etiological agent of CL in Sri Lanka [26]. Sri Lankan *L. donovani* differs from other *L. donovani* strains, at the molecular and biochemical level, showing a distinct genetic association with clinical characteristics in people with CL [26]. Intra-lesional sodium stibogluconate and liquid nitrogen cryotherapy are the primary therapeutic options for CL in Sri Lanka. These treatments are only available in government hospitals with dermatology treatment facilities [27]. Despite the availability of treatments, there is a substantial delay in seeking healthcare for CL among local people, ranging from months to years from the disease onset [28,29]. It is also evident that most CL patients had visited a healthcare professional only when their lesion enlarged, became unusual in appearance, or failed to heal [28,30]. Healthcare-seeking refers to the process of people's thoughts, actions, and behaviours regarding seeking or not seeking treatment for a perceived illness [31]. Symptoms identification and interpretation is the first and foremost step of the healthcare-seeking process [32,33]. People's interpretations of symptoms and associated behaviours are socially constructed [34,35]. Understanding the onset of the disease and the responses of individuals within the context generates knowledge to better inform health authorities in effective disease management, provision of quality healthcare, and development of evidence-based health policies. Qualitative research exploring the perspectives and experiences of people with CL about the onset and symptoms of CL is lacking, both in the global and local context. Therefore, in this study, we aimed a) to explore how individuals describe and interpret the early clinical manifestations of CL and b) how their interpretations and associated behaviours related to early clinical manifestations of CL affected the healthcare-seeking within the socio-cultural context of a rural disease-endemic region in Sri Lanka.

## Methods

### Ethics statement

Ethical approval to conduct this study was obtained from the Ethics Review Committee, Faculty of Medicine and Allied Sciences, Rajarata University of Sri Lanka (ERC/2020/74) and the Faculty of Medicine and Health Sciences, Keele University, United Kingdom [Ref.: MH-

200123]. Informed written consent was obtained from all the participants before data collection. We used a unique participant ID for each participant throughout the data collection, analysis, and presentation. One female participant with poor literacy level was also included in the study. The grandson of the participant provided written consent on behalf of the participant and supported her in completing the study specific data collection instrument.

This study is part of a multi-country (Brazil, Ethiopia, Sri Lanka, and the United Kingdom) global health research programme titled, 'Empowering people with Cutaneous Leishmaniasis: Intervention Programme to improve the patient journey and reduce Stigma via community Education' (ECLIPSE). The ECLIPSE programme includes extensive community engagement and involvement (CEI) to understand community needs, empower communities and turn research findings into effective policies [36].

## Study design

We conducted a qualitative study with thirty people who had completed or were under treatment for CL in three disease-endemic rural communities in the Anuradhapura district, Sri Lanka. Qualitative data were collected from July 2021 to January 2022 through a study-bespoke participant experience reflection journal (PERJ) completed by the participants on their experiences of living with CL and in-depth interviews conducted based on the narratives written in PERJs (post-PERJ interviews).

## Study sites

We selected three study sites from the Anuradhapura district, the largest district in Sri Lanka, with a population of 860 575, with 94.1% of the population residing in rural areas. Paddy cultivation is the main economic activity in the Anuradhapura district [37]. Majority of the population in Anuradhapura district are Sinhalese (91%) in ethnicity and Buddhists (90%) in religion [37]. Between 2001 to 2019, the Anuradhapura district had the highest mean annual incidence of leishmaniasis in Sri Lanka [38], with 344 new CL cases in 2020 [39]. Three local communities within three administrative divisions (Padaviya, Nachchaduwa, and Thalawa) in Anuradhapura district (Fig 1) were purposively selected using the disease prevalence data from the Regional Director of Health Services office, Anuradhapura.

## Participant recruitment

We recruited adults (≥18 years) in the three communities who had completed or were under treatment for CL at a government hospital with CL treatment facilities. Our sample was diverse regarding sex, age, socio-economic and educational backgrounds. Based on the assumption that data saturation could be achieved [40] and the pragmatic considerations of COVID-19 restrictions imposed by the government during the study period, we decided to recruit 30 people with CL residing in three selected study sites with the highest prevalence of CL (10 participants from each study site). We discussed with key community members in each study site (i.e., members from the local farmers' societies, the death benevolent societies [41], youth societies, and women's societies) to identify people with CL in the community. We used the snowball sampling technique for participant recruitment until we reached the expected number and diversity of participants [42].

## Data collection

We used participant experience reflection journals (PERJ) written by people with CL and post-PERJ interviews for data collection.

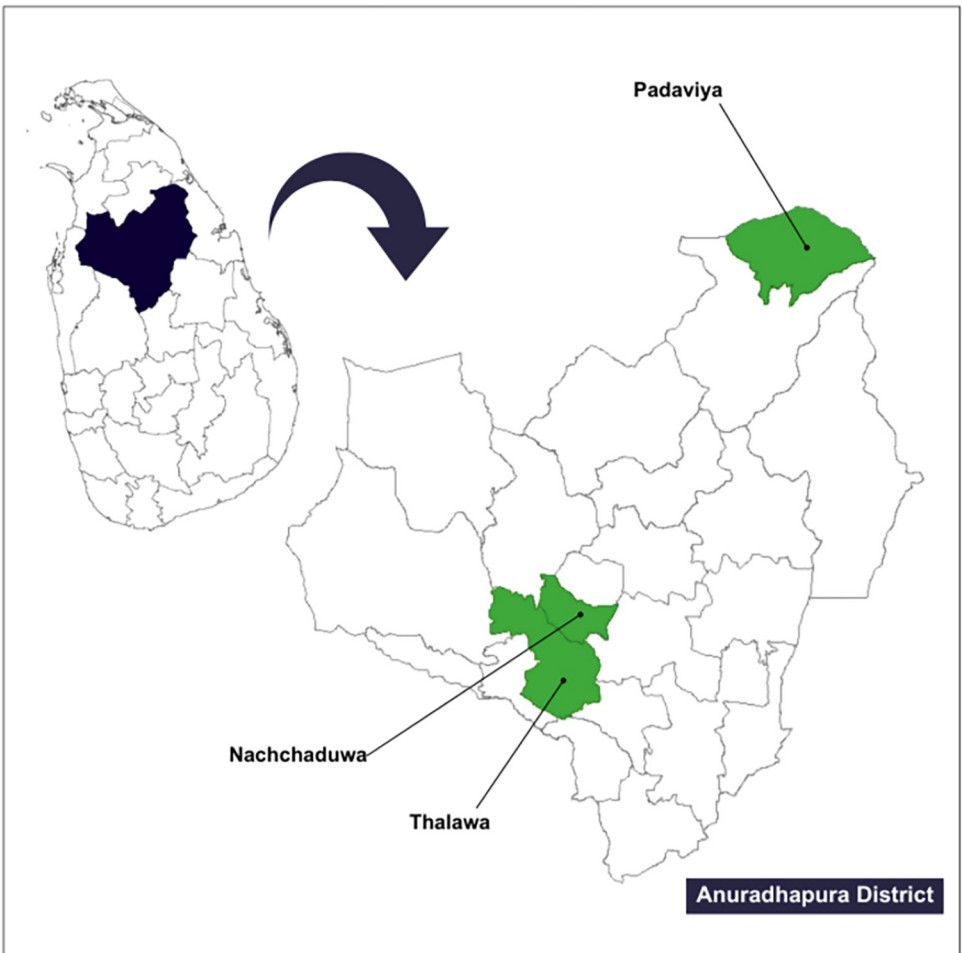

**Fig 1. Study sites located in the administrative divisions in the Anuradhapura district Source of the base file: https://gadm.org/maps/LKA_1.html.**

**a) Participant experience reflection journals.**    Self-reporting is a conventional approach to collect data in health research and is vital for obtaining the perspectives of the study participants [43]. One of the demanding characteristics of self-reporting is participants respond to the questions without the interference of the researcher [43]. We designed a self-reporting PERJ containing 11 open-ended questions. These questions elicit the experiences and reflections of people with CL during different time points of their whole CL patient journey, from the notice of symptoms to recovery (S1 Table). We used published evidence on disease manifestations, implications, and healthcare-seeking in CL and ECLIPSE CEI approach for designing and contextualising the questions in PERJ. We discussed with five community members (with/without CL) about the language used, applicability and appropriateness of the questions to the target group, and format and layout of the PERJ. We incorporated their comments and suggestions to modify the PERJ, and it was reviewed by qualitative research and public health experts in the research team for face and consensual validity. Three ECLIPSE researchers, including the first author of this paper (SDG), collected the data. We met study participants at their residences and explained the study objectives and instructions for completing the PERJ. The period given to complete the PERJ was two to three weeks. We contacted the study

participants once every three/five days via telephone to talk about the PERJ, progress on keeping notes, and clarify any queries.

**b) Post-PERJ interviews.** Assimilated to diary study methods [44] we conducted in-depth interviews with people with CL who wrote the PERJ as the second data collection method. Interviews typically attempt to understand the world from the participant's point of view, reveal the meaning of their experience, and discover their lived world [45]. Based on the data analysis of each PERJ, we identified topics that needed elaboration or clarification and developed a tailored interview guide for each participant. We conducted face-to-face interviews at participants' residences, lasting between 30–120 minutes. Interviewers took field notes during and after each interview, which were conducted in Sinhala and audio-recorded with the participant's permission.

## Data analysis

All handwritten entries were typed to form the PERJ transcripts. Audio-recorded interviews were transcribed ad verbatim and translated from Sinhala into English by research assistants. We followed the narrative-thematic analysis [46], focusing on the chronological sequence of the events from noticing symptoms to the first visit to a modern biomedical healthcare facility. Initially, two ECLIPSE researchers (SDG and TCA) independently coded a sub-set of transcripts and compared the codes generated. Disagreements between the two researchers were then discussed and resolved, and a coding scheme was developed based on the consensus. SDG coded the remaining transcripts according to the coding scheme and added new codes. Consensus on the final coding scheme was reached during research team meetings. We manually coded all the PERJ, and interview transcripts using Sinhala language to preserve the exact meanings and develop meaningful interpretations within the cultural context.

## Research rigour and quality control

The PERJ was designed using CEI approach to develop a context-specific data collection instrument. We used standard guidelines to develop the interview guides and to conduct the post-PERJ interviews [47]. We kept interview memos which later expanded into detailed field notes, to ensure reflexivity [47] and to interpret our study findings. Respondent validation of self-reported data in PERJ was done during the interviews as clarifications, and further elaboration, while maintaining the main focus on gathering additional information related to the written narratives. During the ECLIPSE review meetings conducted at three study sites, we discussed the study findings with community members (with/without CL).

## Results

### Characteristics of the study participants

Thirty people with CL completed the PERJ (12 females and 18 males), of them, 25 were interviewed. The age of the study participants ranged from 18 to 75 years, and the majority were farmers (Table 1). Five participants could not participate in the interview due to the reasons: passing away from another health condition, relocation, being severely unwell, and occupation-related commitments. Two participants were on treatment during the study period, while the rest had already been cured after the treatment.

We identified four major thematic areas during the data analysis: 1) patient-reported early clinical manifestations of CL, 2) local interpretations of the early skin lesion(s) of CL, 3) actions and behaviours in the early stage of CL in the rural context and 4) the time gap between the notice of symptoms and first visit to a modern biomedical healthcare facility.

**Table 1. Characteristics of the study participants.**

| Participant ID | Sex | Age at the time of data collection (Years) | Occupation | Year of notice of CL symptoms | Patient-reported early appearance of the lesion | Location of the lesion(s) | The approximate time gap between the notice of symptoms and the first visit to a biomedical healthcare facility |
|---|---|---|---|---|---|---|---|
| J01 | Female | 55 | Farmer | 2019 | Nodule | Leg | Three months |
| J02 | Female | 27 | Daily wage earner | 2019 | Papule | Leg | 24 months |
| J03 | Male | 24 | Helper in a bakery | 2014 | Papule | Arm | A week |
| J04 | Male | 74 | Retired bank worker | 2008 | Papule | Leg | 12 months |
| J05 | Male | 43 | Salesman | 2009 | Nodule | Leg | 12 months |
| J06 | Female | 31 | Daily wage earner | 2016 | Macule | Leg | 60 months |
| J07 | Female | 32 | Self-employer | 2019 | Papule | Below eye | Two weeks |
| J08 | Male | 26 | Family business | 2015 | Nodule | Behind ear | Six months |
| J09 | Male | 36 | Mason | 2018 | Papule | Arm | Nine months |
| J10 | Female | 41 | Not involved in income generation activity | 2013 | Papule | Leg | A month |
| J11 | Male | 71 | Farmer | 2018 | Eczema | Arm | Three months |
| J12 | Male | 45 | Farmer/Photographer | 2019 | Papule | Abdomen | Six months |
| J13 | Female | 75 | Not involved in income generation activity | 2019 | Papule | Arm Leg | Two months |
| J14 | Male | 56 | Farmer | 2020 | Nodule | Leg | Four months |
| J15 | Female | 54 | Not involved in income generation activity | 2018 | Papule | Finger | < A week |
| J16 | Male | 39 | Driver | 2018 | Papule | Arm | Seven months |
| J17 | Female | 66 | Not involved in income generation activity | 2020 | Papule | Leg | Five months |
| J18 | Male | 73 | Not involved in income generation activity | 2018 | Nodule | Arm | < A week |
| J19 | Male | 44 | Farmer/Carpenter | 2020 | Nodule | Arm Leg | Three months |
| J20 | Male | 32 | Management assistant | 2018 | Papule | Forehead | Seven months |
| J21 | Male | 53 | Farmer | 2019 | Papule | Near ear | Three months |
| J22 | Female | 18 | Not involved in income generation activity | 2018 | Papule | Leg | A month |
| J23 | Male | 60 | Farmer | 2020 | Papule | Arm | A week |
| J24 | Female | 72 | Not involved in income generation activity | 2013 | Macule | Leg | Eight months |
| J25 | Male | 65 | Farmer | 2020 | Papule | Arm | Six months |
| J26 | Male | 32 | Management assistant | 2016 | Nodule | Arm | Three months |
| J27 | Female | 61 | Farmer | 2018 | Papule | Nose | 12 months |
| J28 | Male | 45 | Farmer | 2021 | Papule | Below eye Back Ear | < A week |
| J29 | Male | 50 | Retired soldier | 2017 | Nodule | Leg | 12 months |
| J30 | Female | 47 | School teacher | 2017 | Macule | Leg | < A week |

### Patient-reported early clinical manifestations of CL

**The initial notice of the onset of CL.**    The precise time of CL symptoms onset was not clear to the study participants. The year of the initial notice of CL symptoms by participants ranged between 2008 to 2021, and 22 out of the 30 participants described a disease onset within the last five years (since 2017) (see Table 1). Most participants mentioned that the initial notice of the lesion was an incidental finding that occurred either while working, touching the lesion unintentionally, during bathing, or noticing it in the mirror. A 47-year-old woman described how she accidentally saw and ignored her lesion:

> That did not even hurt. I did not feel it. Of course, I did not even look at it *[Laughs]*. (. . .) I noticed it by accident. Even after noticing it, I did not feel any pain. I thought that it will heal off over time. 'It would have been a bite of a small insect', that is how I thought, a different one, not a sandfly.

> (J30, 47-year-old female, school teacher)

The common reasons for not noticing the lesion at its onset are the non-specific appearance and the absence of pain or sense of CL lesion. But some participants had noticed the lesion due to mild itchiness occurred in the lesion area.

There were instances where another person noticed the lesion. For example, a 61-year-old farming woman revealed that a general practitioner noticed and inquired about a papule on her nose while examining her for another health condition. This was later confirmed as a CL lesion. A nodule-like lesion behind the ear of a 26-year-old man was initially noticed by his barber:

> I do not remember how long I have been with this before going to the doctor. The barber at the saloon was the person who inquired me about it first. He noticed it while giving me a haircut and repeatedly mentioned it every visit. He was used to saying that *'oyāgē kana piṭi-passē geḍiyak tiyanavā. mokakda dannē naē'* (there is a nodule behind your ear, I do not know what it is). That is why I decided to show this to a doctor.

> (J08, 26-year-old male, works in the family business)

**The appearance of the lesion(s) on the initial notice.**    Study participants described the lesion(s) on initial notice by its location, nature, colour, size, and level of pain caused. The area between the knee and ankle, lateral aspects of the arms, and the face (below the eyes, near/ behind ears, on the forehead, and nose) were the sites where the lesions were commonly located (Fig 2). For men, lesions were mostly found on their arms, while for women, it was on the legs. Only two men had lesions on their back and abdomen, and three participants had multiple lesions, which co-occurred or occurred after some time from the first lesion (Table 1).

The most common early clinical manifestation noticed by the participants was a single, small papule (18 participants) or a nodule *(bibilak/gaeṭittak)* (eight participants). Macule-type skin patches were only seen among three women. A 36-year-old mason described the appearance of his initial lesion in PERJ as:

> The papule (*bibila*) looked bigger than a typical mosquito or ant bite. I noticed that it was not going to get better anytime soon. It had a glittery appearance, so I thought it was a pimple and tried to squeeze it.

> (J09, 36-year-old male, mason)

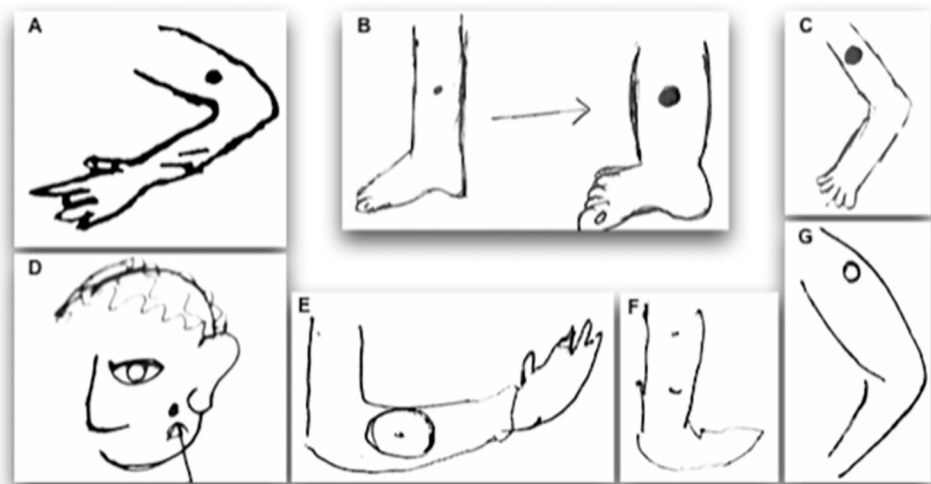

**Fig 2. Sketches of the lesion sites drawn by the study participants in their PERJ.** (A) 24-year-old male (B) 18-year-old female (C) 71-year-old male (D) 53-year-old male (E, F) 44-year-old male (G) 36-year-old male.

Study participants used different Sinhala terms to describe the early appearance of the lesion(s). Each of these terms corresponds with the widely accepted dermatological terms by their meaning (Table 2). When describing the appearance of early lesions, most participants used the phrases 'like a mosquito or an ant bite' or 'like a pimple' as they were common skin manifestations they experienced in day-to-day life. However, some participants used the term 'unusual' to describe the appearance of the lesion(s) due to their suspicious nature.

All the participants identified the lesion when it was small. During interviews, many compared the original size of the lesion to the size of a single seed of dhal (*parippu aeṭayak)*, green gram (*mung aeṭayak*), or turkey berry *(tibbatu gediyak)* to better describe the small size. The original colour of the lesion differed, with the majority being red and some being black, pinkish-white, and yellow. Study participants mentioned contrasting manifestations regarding the level of pain and sensitivity of the lesion. Some participants described that they did not feel any pain and the lesion was non-itchy, while for some, it was painful (two participants) and itchy (nine participants) from the beginning. A 74-year-old retired man described the characteristics of his early lesion as:

**Table 2. Local terms used to describe the early appearance of the lesion(s).**

| The Sinhala term used to describe the lesion by people with CL | Probable dermatological term and the description |
|---|---|
| *bibila* | A papule. It is a circumscribed, elevated, solid lesion that is very small in size. |
| *gaeṭitta/ gaeṭaya/ gediya* | A nodule. These are the areas of abnormally raised skin. These may be hard or soft and small, yet larger than a papule. |
| *lapaya* | A macule. A circumscribed area of discolouration that is small in size. |
| *dadaya* | This term is commonly used to describe eczema-like lesions. It is a condition in which a skin patch becomes itchy, rough, and cracked with or without ulceration. |
| *wisara gedi* | An abscess. A purulent (pus-filled) vesicle. |
| *kurulaēva* | A pimple. |
| *panu lapa* | Small hypo-pigmented patches are believed to be due to gastrointestinal worm infestations. |
| *innek* | A wart. |

Interviewer: So, you said you did not feel anything at the beginning, right?

J18: Yes, I did not. I noticed that nodule (*gediya*) only when it became so itchy

Interviewer: Tell me, how did it look like?

J18: Just like the size of a turkey berry *(tibbatu gediyak)*

(J18, 73-year-old male)

A context-specific feature of describing early manifestations of CL is the use of imagery and metaphors with natural objects from daily life to explain the actual sizes of the lesions.

## Local interpretations of the early skin lesion(s) of CL

Leishmaniasis is known as '*Waeli maessagē leḍē*'/*Waeli maekkagē le ē*' (the disease of the sand-fly/sand flea) among the local people. Before being diagnosed with CL, study participants' awareness of CL was mostly limited to the local name of this disease. However, they had several local interpretations based on the different early clinical manifestations of the lesion, socio-cultural backgrounds and unawareness of the disease.

**A 'normal' skin change.**   Many participants assimilated the early manifestation as a 'normal' skin change ignoring that it could lead to a specific disease. Their reasoning for considering the lesion as 'normal' included the small size, non-visible location, absence of pain, mild itchiness of the lesion, and not turning the early lesion into a wound over time. A 27-year-old daily wage earner described in her PERJ why she considered the symptoms as something 'ordinary':

I did not have any special thoughts on that papule (*bibila*) at first because, at the time, I did not know about the sandfly disease. I thought that this was an ordinary papule. So, I did not seek any treatment for it. I did not feel any pain at all.

(J02, 27-year-old female, daily wage earner)

A 55-year-old female farmer misinterpreted the early manifestation as a change in the skin with ageing, as she had a macule-type lesion at the onset. She further explained that such skin changes are carefully considered only if the affected one is a child.

Interviewer: By then, did you know other villagers also had the sandfly disease?

J01: Yes, villagers knew about this disease [imply they have heard the local name of CL]. But the thing is that we do not care if it is just a dot on our skin, right?' (. . .) You know, we usually do not worry about dots and things that we get on our skin. You see, we do not have the same skin that we had when we were younger. We have black dots and all, right? [Laughs]. So, we do not usually worry about it. We would have if we saw it on a child.

(J01, 55-year-old female farmer)

**An insect bite.**   Many participants assumed that the early lesion was an insect bite caused by mosquitoes or ants due to its appearance. An 18-year-old girl described this in her PERJ:

One day, when I woke up in the morning, I noticed a very small red papule (*bibila*) associated with a hair in the skin on my leg. I thought that this could not be a symptom of a disease. Because the papule was so small and simulated a reddish mosquito bite, I did not pay much attention to it for about 3 or 4 days.

(J22, 18-year-old female)

Experiencing insect bites in their day-to-day life is common, especially in farming communities. This is evident from the following excerpt in a PERJ:

I cannot exactly remember when I got the [sandfly] disease as I have been working on the fields for the past few years. I did not have a proper understanding of the sand-fly disease. We always get insect bites from flies, mosquitoes, and ants when we work in the field. So, getting patches (*lapa*) and wounds like this is normal for us.

(J01, 55-year-old female farmer)

**Other interpretations.**   People with CL had other interpretations of the lesion, such as warts, boils and worm infestations (*panu lapayak*). A 31-year-old woman, pregnant during the onset of CL, suspected that her red-coloured hypo-pigmented patch on the leg could indicate worm infestation as she had missed the worm treatments for several months. This assumption has also been confirmed by the medical officer at her maternal clinic, leading to a five-year delay in receiving treatment for CL.

That's when I was pregnant with my second son in 2016. At that time, it was a small red papule, just like a patch caused by worm infestation. It was not painful or itchy at all. I thought it could be a symptom of worm infestation (*panu lapayak*) as I did not take worm treatment for so long.

(J06, 31-year-old female, daily wage earner)

**Not suspecting the lesions as CL.**   We observed that none of the participants suspected their early lesion was a symptom of CL because (1) interpreting it as common and non-specific skin manifestations exist in these communities based on its appearance, (2) not being aware of CL symptoms, (3) confusion due to atypical appearance of the lesion, and 4) not being aware of the disease CL. Even participants having a close relative with CL have not suspected this as leishmaniasis due to the different manifestations they experienced. A 66-year-old woman explained this when the interviewer referred to her two sons who had CL:

You see, my sons' wounds did not look like mine. They were different. Their ones looked like *Konda Kaewums* (a Sri Lankan sweet meat shaped like a cone with a base and an irregular margin). In mine, the scab fell off and left a flat, dry scar. It was not like that for them. Their one had a different appearance and was protruding up from the skin. I saw first-hand what it looked like with my children. That is why I thought there was no way I too, had the sandfly disease.

(J17, 66-year-old female)

Suspecting the participant's lesion as CL by other people with CL in the area, having relatives/neighbours with CL, becoming aware of CL symptoms through reading, having family members who work in hospitals and self-persuasion by knowing that other villagers visit hospitals to check their skin lesions made the participants suspicious at a later stage.

## Actions and behaviours in the early stage of CL in the rural context

**Ignoring the skin lesions.**   Participants who ignored the early lesion assuming it was a mosquito or an ant bite, as a general minor wound, waited for a few days, weeks, or even

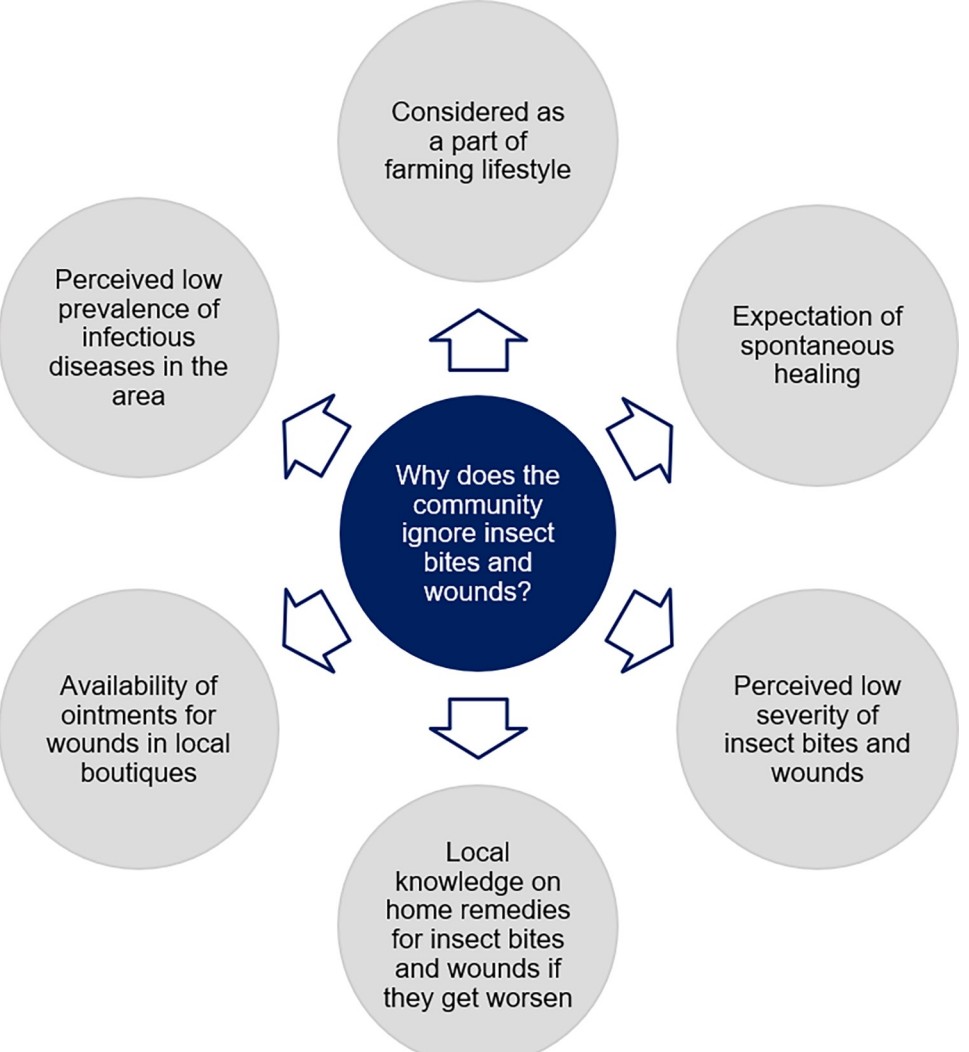

**Fig 3. Reasons for ignoring insect bites and wounds in rural Sri Lanka.**

months expecting spontaneous healing before trying any self-management practice or visiting a healthcare professional.

They explained varied context-specific reasons behind ignoring insect bites and wounds, as outlined in Fig 3.

**Managing wounds at home.**   Participants mentioned that, in general, they self-manage the wounds unless the wound is big or painful, causing disturbances to their daily activities. A 50-year-old retired soldier described this as:

> We do not rush to the hospital for ordinary wounds, right? (. . .) I mean, we go to the hospital only if we cannot manage those by ourselves, in instances when it worsens a little (*Udu dānavā*) and if the wound does not get better.

> (J29, 50-year-old male, retired soldier)

At least one-third of our study participants had initiated self-management of CL symptoms before visiting a healthcare professional.

*Use of different ointments.*    Participants tried several commercially available ointment products to cure CL. These included popular herbal ointment products for wounds, such as *Bilan* and *Vital*, which are cheap and readily available in local boutiques. They mentioned that both ointments are tolerable when working in the water and mud in the paddy fields. This enabled the farmers to continue farming activities, even with a lesion. Apart from the herbal ointment products, some popular ointments used in western medicine, such as *Soframicine* and *Betadine*, have been used by some study participants.

*Use of medicinal plants.*    Some study participants used self-prepared herbal mixtures, including medicinal plants such as *Wel penela* (*Cardiospermum halicacabum*) or *Eththora* (*Senna alata*), for treating CL. Usually, these plant leaves are chopped, placed on the lesion, and tied up using a pure cloth. Because there is a common practice and belief among the local community about using *Wel Penela* for treating abscesses (*wisara gedi*) and *Eththora* for treating eczemas *(dadaya)*.

*Use of Ayurvedic oils.*    Some participants applied an Ayurvedic herbal oil called *Sarwa-vishadi Thailaya* on the CL lesion. This oil is also readily available in local boutiques, and people use it to reduce swelling and pain due to wounds.

None of the participants who used ointments, herbals or oils reported that it reduced the progression of the disease.

*Heating the lesion.*    Three farmers (two men and a woman) described heating the CL lesion using a boiled spoon. They had no specific knowledge about the temperature and time duration for heating. A 60-year-old man and a 61-year-old woman had tried this twice or thrice per week. With advice from another person with CL, a 65-year-old man heated the spoon directly with fire and kept it on the lesion twice a day for two-three days. Some participants think heating the infected area kills the worms inside their lesions. There is a local belief that worms live inside the lesion and the body, which were born from the eggs laid by the sandfly during the bite. All three participants mentioned that the heating could reduce the twitching sensation inside the lesion [believed to be caused by worms].

**Other harmful practices.**    Some participants described the experiences of the other CL patients, who had used harmful practices such as burning the lesion using fire matches, kerosene, and petrol. The exact reasons why those patients used these chemicals were unclear to the study participants.

Apart from these practices, the participants, who interpreted the early lesion as a pimple, had tried to pierce or squeeze it using fingernails or thorns.

**Seeking modern biomedical healthcare.**    Seeking modern biomedical healthcare through the government or private health sectors for CL remained the second option among almost all the study participants. The main reasons for seeking modern biomedical healthcare included 1) the non-healing and long-lasting nature of the lesion, 2) the progression of the lesion over time, 3) the recurring nature of the lesion even after using home remedies, 4) complications that occurred in the timeline of the disease, and 5) advice received from other CL patients in the area. Participants visited the medical officers at the government hospitals (21 participants) or the private medical centres (eight participants). A 74-year-old man described in his PERJ:

> You know warts, do not you? My papule (*bibila*) was also similar to a wart. It naturally fell off. Thinking that it had gotten better, I went to work. But then, after a while, it reappeared. There is a doctor at a private dispensary in the neighbourhood from whom I have sought treatment for other diseases. So, I decided to go there.

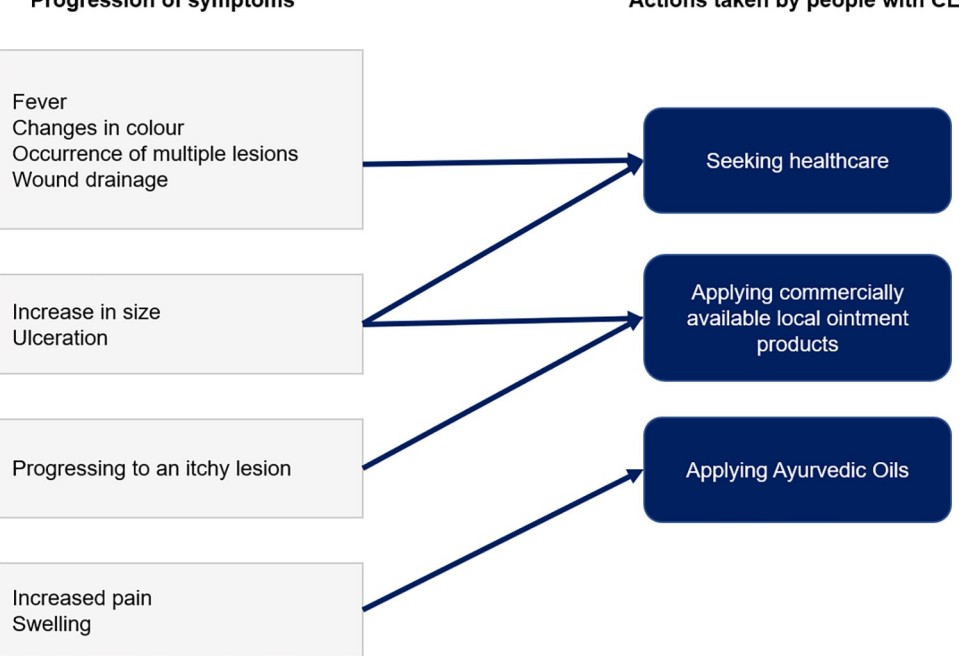

**Fig 4. Progression of early symptoms of CL and actions taken by study participants.**

(J04, 74-year-old male, retired bank worker)

The progression of the lesion has influenced the decision to seek modern biomedical healthcare for CL over time. Fig 4 indicates how the progression of lesions led to take different actions by our study participants.

## The time gap between the notice of symptoms and the first visit to a biomedical healthcare facility

Our focus on the chronology of these events until seeking healthcare gave rise to the patterns of delays generated. People do not seek healthcare immediately after noticing a skin lesion. In the case of CL, the interplay between the people's interpretations, actions, and behaviours could delay the first visit to a healthcare facility (Fig 5).

On average, we observed a substantial time gap (ranging from weeks to years) between the notice of the symptoms of CL and the first visit to a modern biomedical healthcare facility in all three study communities (Table 1). Most participants had visited a biomedical healthcare professional within three to 12 months while few had more than one year delay.

In this excerpt from a PERJ written by a 55-year-old female farmer, she refers to CL as 'a misleading disease' in her explanation of how silently it grows, leading to a delay in healthcare-seeking:

> While days, weeks, and months passed, I wondered whether the sandfly disease is a mis-leading disease (*walimessāgē leḍē rōgīnva nomaňga yavana le ak*). It means that people do not know about this disease. As the wound is not painful, people do not worry about it. They ignore it. They do not even dare to think what it is. And so, the disease gets worse without you knowing it.

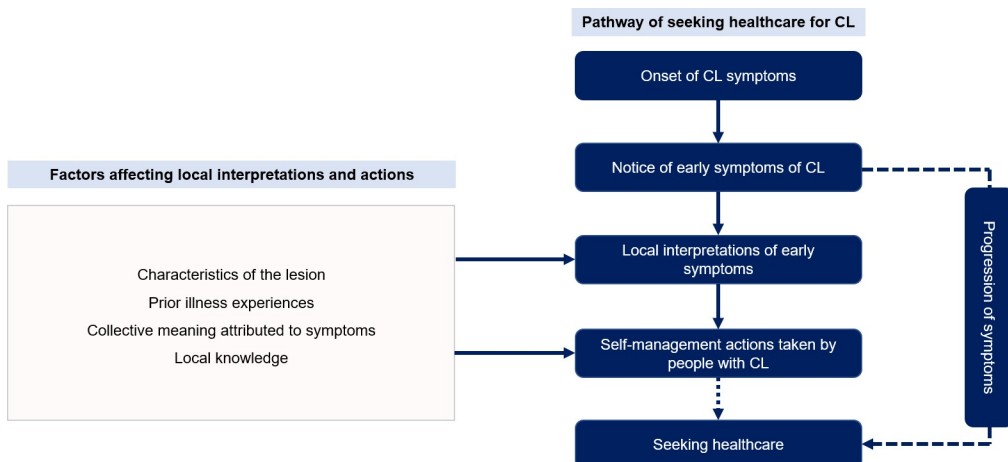

**Fig 5. The interplay between interpretations, actions and behaviours, and seeking healthcare for CL.** As outlined in Fig 5, there are a few potential patterns in seeking healthcare for CL; seeking healthcare just after noticing the CL lesion (is very rare), seeking healthcare with the progression of the symptoms and seeking healthcare after attempting self-management actions to cure CL (these patterns are indicated in dashed lines). People's interpretations, actions and behaviours related to CL clinical manifestations are always affected by the other factors outlined in Fig 5 (light pink-coloured box on the left).

(J01, 55-year-old female farmer)

A delay due to the failure to diagnose CL by general practitioners during the first visit to the biomedical healthcare facility was reported by seven participants. Among them, three participants mentioned two months, five months and, a more than four-year delay in obtaining the correct CL diagnosis.

## Discussion

Our study findings provide insights into local interpretations, actions and behaviours related to early clinical manifestations of CL and their influence on the healthcare-seeking among people with CL in rural Sri Lanka. Patient-reported early CL symptoms had both similarities and several deviations from the standard clinical descriptions of the CL lesions. Most participants had misinterpreted and ignored the early lesion assuming it was a normal skin lesion, mosquito or ant bite, pimple, wart, oil bump, abscess, or a sign of worm infestation. The people with CL perform both harmful and non-harmful behaviours before seeking healthcare. A substantial time gap between the notice of CL symptoms and the first visit to a modern biomedical healthcare facility was observed. This delay is interrelated with the local misinterpretations and associated behaviours of people with CL.

Most studies conducted in Sri Lanka suggest the single, painless, small papule as the typical clinical presentation of CL [30,48]. Even within our study, single papule was the most common manifestation. But variations that mimicked diverse dermatological manifestations, including macules, eczemas, abscesses, and skin patches were also reported. Though non-itchy and non-painful lesion was the typical presentation [30], some study participants had complained about pain and itchiness since noticing the lesion, which is an uncommon finding. Most studies explain the characteristics of the CL lesion(s) based on its appearance when people with CL reach the healthcare setting [29,30,49]. By the time of reaching the healthcare facility, the clinical picture of the early CL lesions could be completely different from its initial appearance. Therefore, our study is important to understand the early CL symptoms experienced by people

with CL, to describe the clinical entity of this disease better and for effective disease management through early diagnosis. The most affected sites by CL are the exposed areas of the body, such as the lower and upper limbs and face [8,28,30,49]. A similar finding has been reported in our study as well. We could identify most of the CL lesions in men on the arms, and in women, it was the legs. Even the farmers and participants who did not engage in income generation activity had lesions mostly on the legs and arms. Two male participants had lesions on the usually unexposed body areas, such as the abdomen and back, similar to a few other studies conducted in Sri Lanka [29,30,49]. This may be closely related to people's clothing habits, which are bound with cultural dimensions [49] and their daily routines. As our study participants live in predominantly agricultural communities, most are engaged in farming paddy, vegetables and fruits and home gardening. Usually, their upper and lower body areas are exposed during farming activities and prone to insect bites making them more vulnerable to acquiring CL.

Our study reveals that different interpretations held by people with CL about the appearance of the early lesion have been more likely to contribute to the delay in healthcare-seeking. A similar finding is evident in a study conducted with people diagnosed with CL at a tertiary referral centre in Colombo, Sri Lanka [28]. Yet, that study does not describe the contextual factors which led to those misinterpretations and people's subsequent responses, which is important in planning health education interventions. Our study fills that gap and describes the underlying context affecting people's interpretations and behaviours related to CL. Our study shows that the participants used local herbal/non-herbal medications for CL based on their easy availability and prior experience of using them for other types of wounds. They mentioned the incidents of burning the lesion with kerosene or petrol by other people with CL in society. A similar practice of using harmful chemicals such as lead, battery acid, and bleach or chlorine to cure CL lesions is commonly observed in countries like Suriname and Ecuador [15,50]. None of these home remedies or chemicals has been medically proven effective for curing CL, but these practices could worsen the lesion and delay treatment seeking. We could observe a similar practice to 'heat therapy', which is defined as 'using heat to treat different health conditions and relieve pain' [51]. There is evidence of the successful medical use of heat therapy for CL around other countries [51]. However, the practice of heating the lesion using a boiled spoon, as reported by the participants of this study, could be a more localised and adjusted version of the standard heat therapy. However, further studies are needed to explore how it is contextualised locally.

Lack of access to healthcare may cause a delay in the diagnosis and treatment of NTDs [52]. In Sri Lanka, treatment for CL is available free of charge in government hospitals where a dermatology clinic is present [53]. Although reaching the facilities may be difficult for rural community members, our study depicts that despite the availability of treatment and the distance to treatment centres from study sites, seeking healthcare for CL remained the second option among the study participants due to various contextual factors described within our study. Therefore, we identify the lack of awareness around CL as a major reason for the delay in healthcare-seeking [54,55]. Understanding the local interpretations and contextual factors that contributed to the choice of performing these actions and behaviours is crucial for empowering people for early healthcare-seeking and bridging the healthcare professionals and local communities in finding ways for effective disease management.

Our study findings provide evidence that there is an urgent need to improve awareness of CL among the general public. Going beyond the typical manifestations of CL, focusing more on the varied manifestations experienced by the participants in this study, could prevent local misinterpretations and failure to diagnose CL by healthcare professionals. The social marketing programme to eliminate leprosy, initiated in the 1990's in Sri Lanka, is one of the best

examples to indicate the importance of improving awareness of early signs and symptoms of skin diseases among the general public and healthcare professionals [56]. The comprehensive understanding of patient-reported early clinical manifestations, local interpretations and behaviours related to CL from this study can be combined with the prior experience of controlling prevalent skin diseases in Sri Lanka to reduce the burden of CL in the future substantially.

There a few anticipated limitations within the study. We aimed to reflect the context-specific comprehensive picture of the patient-reported early CL symptoms and their health-related behaviours rather than attempting to generalise the findings. However, these findings could be used to understand the health-related behaviours underlying other infectious diseases, more broadly, the neglected health issues in similar study populations. Since this study involved recalling past events related to CL, recall bias is possible. However, we have minimised this using validation of information from the same participant using two different data collection methods, asking participants to discuss with their family members to recall events, asking a range of questions from the participant during the post-PERJ interviews and verifying certain information using the participants' clinic book.

A particular strength of our study is the qualitative research design, which provides insights into how the local community interprets and understands CL and its early symptoms, which can be used to design evidence-based, context-specific, and culturally bespoke public health interventions to improve early healthcare-seeking for CL. Also, the findings of this study will be beneficial for creating a platform to break the health communication barriers between healthcare professionals and the local communities for improving early diagnosis. At the national level in Sri Lanka and the international level, the findings would provide the groundwork for health policymakers to understand the behavioural components underlying this neglected health issue. This will ultimately enable the policymakers to contextualise the national and regional level programmes related to leishmaniasis and other infectious diseases for effective service provision and disease control.

## Supporting information

**S1 Table. Content of the participant experience reflection journal (PERJ).**
(PDF)

## Acknowledgments

The authors would like to acknowledge the members of the three ECLIPSE communities for their participation in the research. Also, we thank the Provincial Director of Health Services and Regional Director of Health Services in Anuradhapura, Staff of the Medical Officer of Health areas in Padaviya, Nachchaduwa and Thalawa, Grama Niladhari officers in the selected three study sites for granting permission and necessary support for conducting the study. We extend thanks to our research assistants, Mr. Sandaru Hasaranga Shanthapriya for designing and formatting all the figures included in this manuscript and for transcribing the post-PERJ interviews with Ms. R. Madushika Lakshani Ranathunga.

## Author Contributions

**Conceptualization:** Sonali Dinushika Gunasekara, Nuwan Darshana Wickramasinghe, Suneth Buddhika Agampodi, Manoj Sanjeewa Fernando, Lisa Dikomitis, Thilini Chanchala Agampodi.

**Data curation:** Sonali Dinushika Gunasekara, Thilini Chanchala Agampodi.

**Formal analysis:** Sonali Dinushika Gunasekara, Thilini Chanchala Agampodi.

**Funding acquisition:** Suneth Buddhika Agampodi, Lisa Dikomitis.

**Investigation:** Sonali Dinushika Gunasekara.

**Methodology:** Sonali Dinushika Gunasekara, Nuwan Darshana Wickramasinghe, Suneth Buddhika Agampodi, Kosala Gayan Weerakoon, Lisa Dikomitis, Thilini Chanchala Agampodi.

**Project administration:** Suneth Buddhika Agampodi, Lisa Dikomitis, Thilini Chanchala Agampodi.

**Resources:** Suneth Buddhika Agampodi, Lisa Dikomitis, Thilini Chanchala Agampodi.

**Supervision:** Nuwan Darshana Wickramasinghe, Suneth Buddhika Agampodi, Manoj Sanjeewa Fernando, Lisa Dikomitis, Thilini Chanchala Agampodi.

**Validation:** Sonali Dinushika Gunasekara, Suneth Buddhika Agampodi, Thilini Chanchala Agampodi.

**Visualization:** Sonali Dinushika Gunasekara.

**Writing – original draft:** Sonali Dinushika Gunasekara.

**Writing – review & editing:** Sonali Dinushika Gunasekara, Nuwan Darshana Wickramasinghe, Suneth Buddhika Agampodi, Manoj Sanjeewa Fernando, Kosala Gayan Weerakoon, Chandani Liyanage, Lisa Dikomitis, Thilini Chanchala Agampodi.

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
