## [Decision Letter · Decision Letter 0]

10 Jan 2023

Dear Dr Agampodi

Thank you very much for submitting your manuscript "'We do not rush to the hospital for ordinary wounds (<suḷu tuvāla>)’: Healthcare-seeking in cutaneous leishmaniasis in rural Sri Lanka" for consideration at PLOS Neglected Tropical Diseases. As with all papers reviewed by the journal, your manuscript was reviewed by members of the editorial board and by several independent reviewers. In light of the reviews (below this email), we would like to invite the resubmission of a significantly-revised version that takes into account the reviewers' comments. 

Reviewer #1- Minor revision

General: Gunasekara and co-authors present a study of self-described health seeking behavior related to cutaneous leishmaniasis in Sri Lanka. The manuscript is very well written, clear and logically structured. The topic is relevant as understanding motivations for people to (not) seek health care is crucial for information and education efforts aimed at reducing delays in the process and thereby reduce morbidity and the risk of onward transmission. As with many social science studies, it is unclear how generalizable the insights are beyond the study population and especially the Sinhala Sri Lankan population. It is to be hoped that the group will in the future provide comparative discussions of the results obtained in different countries and thereby be able to better determine which aspects are more universal and which ones rather local culture-dependent.

This reviewer is impressed by the quality of the work and manuscript, and has only few points to offer as specific comments:

- Overall, the manuscript is rather lengthy and the authors are invited to consider shortening it/tightening up to reduce the word count.

- Line 148: what is a “death benevolent society”?

- Line 209 etc.: it is mentioned that 30 participants completed the patient journey booklet. How many potential participants were approached initially?

- Line 253: “the upper area of the hands” – do the authors mean arms rather than hands? “between the shoulder and elbow” seems to refer to a part of the arms

- Line 417 etc.: On multiple occasions, the authors mention the relatively long delay between the moment the participants became aware of the symptoms and the time they sought formal health care. It is unclear whether “seeking care” refers to the first time they presented at a health facility or the moment they received the correct diagnosis? Also, it would be relevant to understand whether there were wrong diagnoses provided first before the final diagnosis of leishmaniasis had been established? In other words: How long was the delay between seeking health care for the first time and receiving the correct diagnosis and how common are wrong initial diagnoses (some comments seem to imply they exist)?

Reviewer #2 – Major revision

This is a strong paper about an important topic. The research methodology is strong, although some more details would be helpful. The data itself is rich and well presented. Further interpreting and organizing the findings, whether in line with a theoretical framework about healthcare seeking behavior, or even using a barriers and facilitators to care approach would deepen the potential to draw conclusions. Finally, if possible, including a socio-economic and equity analysis would round out the interpretation within the context, which currently addresses primarily culture but not other contextual influences.

I- Methods

- The qualitative research design is strong – my overall recommendation is to start with one or two sentences summarizing the approach, to let the reader know what they are going to read details about.

- The objectives of the research with a clear testable hypothesis are not stated within the methods section. The objective of exploring individual perceptions of early CL manifestations and health-related behaviors, within the rural Sri Lankan context, is at the end of the introduction. This could instead be expanded into two or three clear objectives within the methods section.

- Study design section could benefit from more detail. Reorganizing the participant recruitment section could be helpful – first describing what people were eligible, how they were identified (perhaps in more detail than is currently presented), and then the criteria used to select among them. Think about giving enough detail so someone replicate the process. Similarly, describe all methods related to booklets and all methods related to interviews, or add subheadings to differentiate between the two data collection methodologies.

- Sample size – appropriate for a qualitative study like this

- Codebook development and application is appropriate according to the methodology described. Please clarify if both coders each coded every transcript? I imagine that’s what you mean, but from the text it isn’t clear.

- I would describe the participatory approach used to develop the research and data collection tools earlier in the manuscript. It’s important from an equity perspective and crucial to the credibility and reliability of the data collected.

- How were field notes used during analysis, if at all? Or were they soley developed to ensure reflexivity?

- Were findings shared with members of the CL community to validate findings and ensure responses were interpreted correctly? This is often considered a crucial step in qualitative research and is worth describing if it was done.

- No concerns about ethical or regulatory requirements. Add a description of how patients with limited literacy were consented, given that the methods says they were included.

II- Results

It would be helpful for interpretation of quotations (which are great!) to have a sentence before to introduce them, then a sentence after each quote to guide the reader in interpreting and integrating the information.

Concerns about the length of time that has passed? Recall bias for events so far in the past can be significant, should be included in the limiatations.

Figure 3 has important information, but could throw off the reader because the manner of presenting similar data shifts. Consistency in data presentation, whether via text, tables, or figures, could be helpful.

Subdividing the long paragraph on treatments attempted by patients would be helpful, perhaps dividing between Western medicine and traditional medicine? Also mentioning if any of these treatments can be effective would be helpful for people not very familiar with CL.

Table 3 might be visualized another way to connect the data – perhaps showing which types of progression led to which actions (a flow chart perhaps, or some way to show that the second column contains many repeating actions)

Figure 4 is great conceptually and definitely needed to bring findings together, but a little confusing to interpret. Adding some labels and explaining what the different arrows mean and why they point to different boxes would be helpful.

There are also conceptual frameworks related to healthcare seeking behavior that could be useful in the presentation and interpretation of findings.

III- Conclusions

Conclusions are supported by the data presented, but further analysis and interpretation could be useful.

Reasons behind the delay in care seeking – did you explore socio-economic reasons for delay? Such as distance or cost to receive care? Distrust of healthcare facilities? A further analysis, even in the discussion, that includes an equity of healthcare access and quality of care received are important. For example, from the data you presented it seems like a priority was returning to work, did this need to keep working influence care access? Framing this around barriers to and facilitators of care access could be another framing approach.

Authors discussed how data can be used to inform practice and policy.

Limitations section was not present and would be valuable.

- Much of the manuscript might benefit from a bit of reorganization, ensuring like information is with like. For example, in the Introduction, describing transmission, then manifestations, then consequences of infection, then reasons to prioritize early diagnosis. Outlining then grouping information together by theme could help reduce the minor confusion when currently reading it.

- Adding details to the introduction about prevalence would be helpful, prevalence is described as “high” compared to other countries in the Asian subcontinent, numbers would be useful.

- Tables and figures are needed and useful, but none add clarity to interpretation, which is often the hope with these tools, but instead require their own interpretation. Making them clearer would be a huge help to the reader.

Reviewer # 3- Major revision

This is very important research to better understand and develop site specific interventions to improve early detection and treatment for CL. 

Availability of data- clarify how one can access this data.

I- Abstract:

- line 25- "failure" to understand. Failure is a strong word and failure of who or what. Reconsider alternative wording.

- findings- include a sentence on time gap

The study design and methods are presented clearly and are well suited to address the aim of the research.

- it is not clear what pool of eligible pts were identified prior to purposively selecting 30 adults. How many CL past/current patients did the community help identify?

- include how you accommodated illiterate pts under 'ethical considerations'

II – Results

The results are presented based on the 4 themes. Different methods of presenting the data are much appreciated- good choice of quotes, excerpts from booklets, and tables.

- line 224- "time" of consent of CL may not be the best term since no one can pin point a time of onset. The results presented are more about how and what they noticed. I would reconsider the use of 'time'. Possibly using 'awareness of onset of CL'?

- "a symptom of worm infestation" (line 309) and "a change due to ageing" (line 321)- give some weight to how often these sub themes emerged. It appears to be the experience of only one pt for each. If they are only one consider a sub theme that covers a few of these 'other' experiences. "A change due to ageing" sounds like it fits under "a normal skin change"- consider describing this case under that sub-theme.

- "non-suspicion due to lack of awareness" (line 335)- This section has some information that could benefit being presented early in this section. Readers want to know earlier what the local term for CL is, its meaning and how is CL understood (or not) in this context. Consider moving before "a normal skin change"

- in line with the previous comment, you use the term 'sandfly disease' a few times and its not clear how much of this is part of the local understanding. Having this background gives some context to their health seeking choices. In line 327 it is said that "villagers knew about the disease"- how and what do they know? if that knowledge is there, the results do not discuss any pts that may have then suspected it was CL or sandfly disease. Line 337 says "most of the pts did not suspect their lesion to be CL ... " This implies that some/very few may have suspected it as CL/sandfly disease. Further at line 391, beliefs about eggs laid by the sandfly during the bite is part of the local understanding of CL/sandflly disease but again data is not presented on how, when, why any pts consider their symptoms as potentially being CL and if this prompts them seek formal health care. This needs clarity.

- line 382- "used herbal mixtures prepared using medicinal plants"- if you have the data to briefly provide more context around who and/or the process of preparing herbal mixtures- is it done by pts themselves or do they go to an herbalist or shop?

- line 407- is an Ayurvedic hospital considered formal health care? If so add at line 188 where you describe formal healthcare. But this definition at line 188 defines it as health facilities that have western medicine which I question if ayurvedic hospitals do.

- line 401- correct "all most" to almost

III- Conclusions

- include limitations to the study

- line 500- delete 'for' in affected for people's

- line 535- 'how local knowledge is produced'- As stated in results section, this did not come out clearly in the results.

- include point 3 of recommendations in the author summary in the main paper recommendations. Its an important recommendation.

We cannot make any decision about publication until we have seen the revised manuscript and your response to the reviewers' comments. Your revised manuscript is also likely to be sent to reviewers for further evaluation.

Sincerely,

Manoel Otávio Costa Rocha

Academic Editor

Charles Jaffe

Section Editor

Dear authors,

The article was submitted to analysis by three independent reviewers, whose main recommendations can be found below. After due consideration of the recommendations, the article may be forwarded for further publication review.

Yours sincerely.

The Editorial Board

Reviewer #1- Minor revision

General: Gunasekara and co-authors present a study of self-described health seeking behavior related to cutaneous leishmaniasis in Sri Lanka. The manuscript is very well written, clear and logically structured. The topic is relevant as understanding motivations for people to (not) seek health care is crucial for information and education efforts aimed at reducing delays in the process and thereby reduce morbidity and the risk of onward transmission. As with many social science studies, it is unclear how generalizable the insights are beyond the study population and especially the Sinhala Sri Lankan population. It is to be hoped that the group will in the future provide comparative discussions of the results obtained in different countries and thereby be able to better determine which aspects are more universal and which ones rather local culture-dependent.

This reviewer is impressed by the quality of the work and manuscript, and has only few points to offer as specific comments:

- Overall, the manuscript is rather lengthy and the authors are invited to consider shortening it/tightening up to reduce the word count.

- Line 148: what is a “death benevolent society”?

- Line 209 etc.: it is mentioned that 30 participants completed the patient journey booklet. How many potential participants were approached initially?

- Line 253: “the upper area of the hands” – do the authors mean arms rather than hands? “between the shoulder and elbow” seems to refer to a part of the arms

- Line 417 etc.: On multiple occasions, the authors mention the relatively long delay between the moment the participants became aware of the symptoms and the time they sought formal health care. It is unclear whether “seeking care” refers to the first time they presented at a health facility or the moment they received the correct diagnosis? Also, it would be relevant to understand whether there were wrong diagnoses provided first before the final diagnosis of leishmaniasis had been established? In other words: How long was the delay between seeking health care for the first time and receiving the correct diagnosis and how common are wrong initial diagnoses (some comments seem to imply they exist)?

Reviewer #2 – Major revision

This is a strong paper about an important topic. The research methodology is strong, although some more details would be helpful. The data itself is rich and well presented. Further interpreting and organizing the findings, whether in line with a theoretical framework about healthcare seeking behavior, or even using a barriers and facilitators to care approach would deepen the potential to draw conclusions. Finally, if possible, including a socio-economic and equity analysis would round out the interpretation within the context, which currently addresses primarily culture but not other contextual influences.

I- Methods

- The qualitative research design is strong – my overall recommendation is to start with one or two sentences summarizing the approach, to let the reader know what they are going to read details about.

- The objectives of the research with a clear testable hypothesis are not stated within the methods section. The objective of exploring individual perceptions of early CL manifestations and health-related behaviors, within the rural Sri Lankan context, is at the end of the introduction. This could instead be expanded into two or three clear objectives within the methods section.

- Study design section could benefit from more detail. Reorganizing the participant recruitment section could be helpful – first describing what people were eligible, how they were identified (perhaps in more detail than is currently presented), and then the criteria used to select among them. Think about giving enough detail so someone replicate the process. Similarly, describe all methods related to booklets and all methods related to interviews, or add subheadings to differentiate between the two data collection methodologies.

- Sample size – appropriate for a qualitative study like this

- Codebook development and application is appropriate according to the methodology described. Please clarify if both coders each coded every transcript? I imagine that’s what you mean, but from the text it isn’t clear.

- I would describe the participatory approach used to develop the research and data collection tools earlier in the manuscript. It’s important from an equity perspective and crucial to the credibility and reliability of the data collected.

- How were field notes used during analysis, if at all? Or were they soley developed to ensure reflexivity?

- Were findings shared with members of the CL community to validate findings and ensure responses were interpreted correctly? This is often considered a crucial step in qualitative research and is worth describing if it was done.

- No concerns about ethical or regulatory requirements. Add a description of how patients with limited literacy were consented, given that the methods says they were included.

II- Results

It would be helpful for interpretation of quotations (which are great!) to have a sentence before to introduce them, then a sentence after each quote to guide the reader in interpreting and integrating the information.

Concerns about the length of time that has passed? Recall bias for events so far in the past can be significant, should be included in the limiatations.

Figure 3 has important information, but could throw off the reader because the manner of presenting similar data shifts. Consistency in data presentation, whether via text, tables, or figures, could be helpful.

Subdividing the long paragraph on treatments attempted by patients would be helpful, perhaps dividing between Western medicine and traditional medicine? Also mentioning if any of these treatments can be effective would be helpful for people not very familiar with CL.

Table 3 might be visualized another way to connect the data – perhaps showing which types of progression led to which actions (a flow chart perhaps, or some way to show that the second column contains many repeating actions)

Figure 4 is great conceptually and definitely needed to bring findings together, but a little confusing to interpret. Adding some labels and explaining what the different arrows mean and why they point to different boxes would be helpful.

There are also conceptual frameworks related to healthcare seeking behavior that could be useful in the presentation and interpretation of findings.

III- Conclusions

Conclusions are supported by the data presented, but further analysis and interpretation could be useful.

Reasons behind the delay in care seeking – did you explore socio-economic reasons for delay? Such as distance or cost to receive care? Distrust of healthcare facilities? A further analysis, even in the discussion, that includes an equity of healthcare access and quality of care received are important. For example, from the data you presented it seems like a priority was returning to work, did this need to keep working influence care access? Framing this around barriers to and facilitators of care access could be another framing approach.

Authors discussed how data can be used to inform practice and policy.

Limitations section was not present and would be valuable.

- Much of the manuscript might benefit from a bit of reorganization, ensuring like information is with like. For example, in the Introduction, describing transmission, then manifestations, then consequences of infection, then reasons to prioritize early diagnosis. Outlining then grouping information together by theme could help reduce the minor confusion when currently reading it.

- Adding details to the introduction about prevalence would be helpful, prevalence is described as “high” compared to other countries in the Asian subcontinent, numbers would be useful.

- Tables and figures are needed and useful, but none add clarity to interpretation, which is often the hope with these tools, but instead require their own interpretation. Making them clearer would be a huge help to the reader.

Reviewer # 3- Major revision

This is very important research to better understand and develop site specific interventions to improve early detection and treatment for CL. 

Availability of data- clarify how one can access this data.

I- Abstract:

- line 25- "failure" to understand. Failure is a strong word and failure of who or what. Reconsider alternative wording.

- findings- include a sentence on time gap

The study design and methods are presented clearly and are well suited to address the aim of the research.

- it is not clear what pool of eligible pts were identified prior to purposively selecting 30 adults. How many CL past/current patients did the community help identify?

- include how you accommodated illiterate pts under 'ethical considerations'

II – Results

The results are presented based on the 4 themes. Different methods of presenting the data are much appreciated- good choice of quotes, excerpts from booklets, and tables.

- line 224- "time" of consent of CL may not be the best term since no one can pin point a time of onset. The results presented are more about how and what they noticed. I would reconsider the use of 'time'. Possibly using 'awareness of onset of CL'?

- "a symptom of worm infestation" (line 309) and "a change due to ageing" (line 321)- give some weight to how often these sub themes emerged. It appears to be the experience of only one pt for each. If they are only one consider a sub theme that covers a few of these 'other' experiences. "A change due to ageing" sounds like it fits under "a normal skin change"- consider describing this case under that sub-theme.

- "non-suspicion due to lack of awareness" (line 335)- This section has some information that could benefit being presented early in this section. Readers want to know earlier what the local term for CL is, its meaning and how is CL understood (or not) in this context. Consider moving before "a normal skin change"

- in line with the previous comment, you use the term 'sandfly disease' a few times and its not clear how much of this is part of the local understanding. Having this background gives some context to their health seeking choices. In line 327 it is said that "villagers knew about the disease"- how and what do they know? if that knowledge is there, the results do not discuss any pts that may have then suspected it was CL or sandfly disease. Line 337 says "most of the pts did not suspect their lesion to be CL ... " This implies that some/very few may have suspected it as CL/sandfly disease. Further at line 391, beliefs about eggs laid by the sandfly during the bite is part of the local understanding of CL/sandflly disease but again data is not presented on how, when, why any pts consider their symptoms as potentially being CL and if this prompts them seek formal health care. This needs clarity.

- line 382- "used herbal mixtures prepared using medicinal plants"- if you have the data to briefly provide more context around who and/or the process of preparing herbal mixtures- is it done by pts themselves or do they go to an herbalist or shop?

- line 407- is an Ayurvedic hospital considered formal health care? If so add at line 188 where you describe formal healthcare. But this definition at line 188 defines it as health facilities that have western medicine which I question if ayurvedic hospitals do.

- line 401- correct "all most" to almost

III- Conclusions

- include limitations to the study

- line 500- delete 'for' in affected for people's

- line 535- 'how local knowledge is produced'- As stated in results section, this did not come out clearly in the results.

- include point 3 of recommendations in the author summary in the main paper recommendations. Its an important recommendation.

Reviewer's Responses to Questions

**Key Review Criteria Required for Acceptance?**

**Methods**

-Are the objectives of the study clearly articulated with a clear testable hypothesis stated?

-Is the study design appropriate to address the stated objectives?

-Is the population clearly described and appropriate for the hypothesis being tested?

-Is the sample size sufficient to ensure adequate power to address the hypothesis being tested?

-Were correct statistical analysis used to support conclusions?

-Are there concerns about ethical or regulatory requirements being met?

Reviewer #1: No concerns

Reviewer #2: - The qualitative research design is strong – my overall recommendation is to start with one or two sentences summarizing the approach, to let the reader know what they are going to read details about.

- The objectives of the research with a clear testable hypothesis are not stated within the methods section. The objective of exploring individual perceptions of early CL manifestations and health-related behaviors, within the rural Sri Lankan context, is at the end of the introduction. This could instead be expanded into two or three clear objectives within the methods section.

- Study design section could benefit from more detail. Reorganizing the participant recruitment section could be helpful – first describing what people were eligible, how they were identified (perhaps in more detail than is currently presented), and then the criteria used to select among them. Think about giving enough detail so someone replicate the process. Similarly, describe all methods related to booklets and all methods related to interviews, or add subheadings to differentiate between the two data collection methodologies.

- Sample size – appropriate for a qualitative study like this

- Codebook development and application is appropriate according to the methodology described. Please clarify if both coders each coded every transcript? I imagine that’s what you mean, but from the text it isn’t clear. 

- I would describe the participatory approach used to develop the research and data collection tools earlier in the manuscript. It’s important from an equity perspective and crucial to the credibility and reliability of the data collected.

- How were field notes used during analysis, if at all? Or were they soley developed to ensure reflexivity?

- Were findings shared with members of the CL community to validate findings and ensure responses were interpreted correctly? This is often considered a crucial step in qualitative research and is worth describing if it was done.

- No concerns about ethical or regulatory requirements. Add a description of how patients with limited literacy were consented, given that the methods says they were included.

Reviewer #3: Abstract:

- line 25- "failure" to understand. Failure is a strong word and failure of who or what. Reconsider alternative wording. 

- findings- include a sentence on time gap

The study design and methods are presented clearly and are well suited to address the aim of the research. 

- it is not clear what pool of eligible pts were identified prior to purposively selecting 30 adults. How many CL past/current patients did the community help identify?

- include how you accommodated illiterate pts under 'ethical considerations'

**Results**

-Does the analysis presented match the analysis plan?

-Are the results clearly and completely presented?

-Are the figures (Tables, Images) of sufficient quality for clarity?

Reviewer #1: see below

Reviewer #2: It would be helpful for interpretation of quotations (which are great!) to have a sentence before to introduce them, then a sentence after each quote to guide the reader in interpreting and integrating the information.

Concerns about the length of time that has passed? Recall bias for events so far in the past can be significant, should be included in the limiatations.

Figure 3 has important information, but could throw off the reader because the manner of presenting similar data shifts. Consistency in data presentation, whether via text, tables, or figures, could be helpful. 

Subdividing the long paragraph on treatments attempted by patients would be helpful, perhaps dividing between Western medicine and traditional medicine? Also mentioning if any of these treatments can be effective would be helpful for people not very familiar with CL.

Table 3 might be visualized another way to connect the data – perhaps showing which types of progression led to which actions (a flow chart perhaps, or some way to show that the second column contains many repeating actions)

Figure 4 is great conceptually and definitely needed to bring findings together, but a little confusing to interpret. Adding some labels and explaining what the different arrows mean and why they point to different boxes would be helpful.

There are also conceptual frameworks related to healthcare seeking behavior that could be useful in the presentation and interpretation of findings.

Reviewer #3: The results are presented based on the 4 themes. Different methods of presenting the data are much appreciated- good choice of quotes, excerpts from booklets, and tables. 

- line 224- "time" of consent of CL may not be the best term since no one can pin point a time of onset. The results presented are more about how and what they noticed. I would reconsider the use of 'time'. Possibly using 'awareness of onset of CL'?

- "a symptom of worm infestation" (line 309) and "a change due to ageing" (line 321)- give some weight to how often these sub themes emerged. It appears to be the experience of only one pt for each. If they are only one consider a sub theme that covers a few of these 'other' experiences. "A change due to ageing" sounds like it fits under "a normal skin change"- consider describing this case under that sub-theme.

- "non-suspicion due to lack of awareness" (line 335)- This section has some information that could benefit being presented early in this section. Readers want to know earlier what the local term for CL is, its meaning and how is CL understood (or not) in this context. Consider moving before "a normal skin change"

- in line with the previous comment, you use the term 'sandfly disease' a few times and its not clear how much of this is part of the local understanding. Having this background gives some context to their health seeking choices. In line 327 it is said that "villagers knew about the disease"- how and what do they know? if that knowledge is there, the results do not discuss any pts that may have then suspected it was CL or sandfly disease. Line 337 says "most of the pts did not suspect their lesion to be CL ... " This implies that some/very few may have suspected it as CL/sandfly disease. Further at line 391, beliefs about eggs laid by the sandfly during the bite is part of the local understanding of CL/sandflly disease but again data is not presented on how, when, why any pts consider their symptoms as potentially being CL and if this prompts them seek formal health care. This needs clarity. 

- line 382- "used herbal mixtures prepared using medicinal plants"- if you have the data to briefly provide more context around who and/or the process of preparing herbal mixtures- is it done by pts themselves or do they go to an herbalist or shop?

- line 407- is an Ayurvedic hospital considered formal health care? If so add at line 188 where you describe formal healthcare. But this definition at line 188 defines it as health facilities that have western medicine which I question if ayurvedic hospitals do. 

- line 401- correct "all most" to almost

**Conclusions**

-Are the conclusions supported by the data presented?

-Are the limitations of analysis clearly described?

-Do the authors discuss how these data can be helpful to advance our understanding of the topic under study?

-Is public health relevance addressed?

Reviewer #1: no concerns

Reviewer #2: Conclusions are supported by the data presented, but further analysis and interpretation could be useful. 

Reasons behind the delay in care seeking – did you explore socio-economic reasons for delay? Such as distance or cost to receive care? Distrust of healthcare facilities? A further analysis, even in the discussion, that includes an equity of healthcare access and quality of care received are important. For example, from the data you presented it seems like a priority was returning to work, did this need to keep working influence care access? Framing this around barriers to and facilitators of care access could be another framing approach. 

Authors discussed how data can be used to inform practice and policy.

Limitations section was not present and would be valuable.

Reviewer #3: - include limitations to the study

- line 500- delete 'for' in affected for people's

- line 535- 'how local knowledge is produced'- As stated in results section, this did not come out clearly in the results.

- include point 3 of recommendations in the author summary in the main paper recommendations. Its an important recommendation.

**Editorial and Data Presentation Modifications?**

Reviewer #1: none

Reviewer #2: - Copy editing would facilitate reader comprehension

- Much of the manuscript might benefit from a bit of reorganization, ensuring like information is with like. For example, in the Introduction, describing transmission, then manifestations, then consequences of infection, then reasons to prioritize early diagnosis. Outlining then grouping information together by theme could help reduce the minor confusion when currently reading it.

- Adding details to the introduction about prevalence would be helpful, prevalence is described as “high” compared to other countries in the Asian subcontinent, numbers would be useful.

- Tables and figures are needed and useful, but none add clarity to interpretation, which is often the hope with these tools, but instead require their own interpretation. Making them clearer would be a huge help to the reader.

Reviewer #3: (No Response)

**Summary and General Comments**

Reviewer #1: General: Gunasekara and co-authors present a study of self-described health seeking behavior related to cutaneous leishmaniasis in Sri Lanka. The manuscript is very well written, clear and logically structured. The topic is relevant as understanding motivations for people to (not) seek health care is crucial for information and education efforts aimed at reducing delays in the process and thereby reduce morbidity and the risk of onward transmission. As with many social science studies, it is unclear how generalizable the insights are beyond the study population and especially the Sinhala Sri Lankan population. It is to be hoped that the group will in the future provide comparative discussions of the results obtained in different countries and thereby be able to better determine which aspects are more universal and which ones rather local culture-dependent. 

This reviewer is impressed by the quality of the work and manuscript, and has only few points to offer as specific comments:

- Overall, the manuscript is rather lengthy and the authors are invited to consider shortening it/tightening up to reduce the word count. 

- Line 148: what is a “death benevolent society”?

- Line 209 etc.: it is mentioned that 30 participants completed the patient journey booklet. How many potential participants were approached initially? 

- Line 253: “the upper area of the hands” – do the authors mean arms rather than hands? “between the shoulder and elbow” seems to refer to a part of the arms

- Line 417 etc.: On multiple occasions, the authors mention the relatively long delay between the moment the participants became aware of the symptoms and the time they sought formal health care. It is unclear whether “seeking care” refers to the first time they presented at a health facility or the moment they received the correct diagnosis? Also, it would be relevant to understand whether there were wrong diagnoses provided first before the final diagnosis of leishmaniasis had been established? In other words: How long was the delay between seeking health care for the first time and receiving the correct diagnosis and how common are wrong initial diagnoses (some comments seem to imply they exist)?

Reviewer #2: This is a strong paper about an important topic. The research methodology is strong, although some more details would be helpful. The data itself is rich and well presented. Further interpreting and organizing the findings, whether in line with a theoretical framework about healthcare seeking behavior, or even using a barriers and facilitators to care approach would deepen the potential to draw conclusions. Finally, if possible, including a socio-economic and equity analysis would round out the interpretation within the context, which currently addresses primarily culture but not other contextual influences.

Reviewer #3: This is very important research to better understand and develop site specific interventions to improve early detection and treatment for CL. 

Availability of data- clarify how one can access this data.

PLOS authors have the option to publish the peer review history of their article (what does this mean?). If published, this will include your full peer review and any attached files.

Reviewer #1: No

Reviewer #2: No

Reviewer #3: Yes: Tara B Mtuy
---

## [Editor Report · Decision Letter 1]

29 Mar 2023

Dear Prof Agampodi,

Thank you very much for submitting your manuscript "' We do not rush to the hospital for ordinary wounds (suḷu tuvāla )' : A qualitative study on the early clinical manifestations of cutaneous leishmaniasis  and associated health behaviours in rural Sri Lanka" for consideration at PLOS Neglected Tropical Diseases. As with all papers reviewed by the journal, your manuscript was reviewed by members of the editorial board and by several independent reviewers. In light of the reviews (below this email), we would like to invite the resubmission of a significantly-revised version that takes into account the reviewers' comments. 

We cannot make any decision about publication until we have seen the revised manuscript and your response to the reviewers' comments. Your revised manuscript is also likely to be sent to reviewers for further evaluation.

Sincerely,

Manoel Otávio Costa Rocha

Academic Editor

Charles Jaffe

Section Editor
---

## [Editor Report · Decision Letter 2]

29 Apr 2023

Dear Dr Agampodi

We are pleased to inform you that your manuscript '' We do not rush to the hospital for ordinary wounds (suḷu tuvāla )' : A qualitative study on the early clinical manifestations of cutaneous leishmaniasis  and associated health behaviours in rural Sri Lanka' has been provisionally accepted for publication in PLOS Neglected Tropical Diseases.

Best regards,

Manoel Otávio Costa Rocha

Academic Editor

Charles Jaffe

Section Editor

The authors satisfactorily answered the questions raised by the reviewers and made several changes that resulted in a significant improvement of the manuscript. Therefore, I am in favor of approving the work in the revised format.

<quillbot-extension-portal></quillbot-extension-portal><quillbot-extension-portal></quillbot-extension-portal>

---

## [Editor Report · Acceptance letter]

10 May 2023

Dear Prof Agampodi,

We are delighted to inform you that your manuscript, "' We do not rush to the hospital for ordinary wounds (*suḷu tuvāla*)' : A qualitative study on the early clinical manifestations of cutaneous leishmaniasis  and associated health behaviours in rural Sri Lanka," has been formally accepted for publication in PLOS Neglected Tropical Diseases.

Best regards,

Shaden Kamhawi

co-Editor-in-Chief

Paul Brindley

co-Editor-in-Chief
